# Fungal Extracellular Enzymes from *Aspergillus* spp. as Promising Candidates for Extra-Heavy Oil Degradation and Enhanced Oil Recovery

**DOI:** 10.3390/microorganisms12112248

**Published:** 2024-11-07

**Authors:** Junhui Zhang, Wendi Feng, Lu Ren

**Affiliations:** 1College of Ecology and Environment, Key Laboratory of Oasis Ecology of the Ministry of Education, Xinjiang University, Urumqi 830046, China; fengwendi0985@163.com; 2Xinjiang Oilfield Company, PetroChina, Karamay 834000, China; renlu834000@163.com

**Keywords:** extra-heavy crude oil, microbial-enhanced oil recovery, fungal enzyme, enzymatic degradation, oil-based bioproducts

## Abstract

**Highlights:**

**Abstract:**

Heavy crude oil (HCO) and extra-heavy crude oil (EHCO) with high viscosity and density pose enormous challenges to the exploitation of oil reserves. While bacteria are increasingly used in biocatalytic upgrading of HCO and EHCO, less attention has been paid to the potential of fungi. The aim of this study was to ascertain the role of fungal extracellular enzymes from *Aspergillus* spp. In the biodegradation of EHCO and their application potential for enhanced oil recovery. *A. terreus* HJ2 and *A. nidulans* HJ4 with the ability to biodegrade HCO were previously isolated from bitumen enrichment cultures. Both strains grew well on EHCO agar plates supplemented with a small amount of soluble starch (0.2%) and yeast extract (0.3%). Extracellular enzymes from each strain separately, as well as mixtures of the enzymes, exhibited EHCO degradation activity, leading to redistribution of hydrocarbons with substantial formation of biogases and organic acids in a 7-day period. Enzymatic degradation resulted in decreased contents of resins and asphaltenes, accompanied by increased contents of saturates and aromatics. Gas chromatography–mass spectrometry revealed distinct redistribution patterns of *n*-alkane in the biotreated oil. Enzymatic degradation additionally caused considerable reduction in oil viscosity (by 12.7%) and heavy metal concentrations (Ni, by 44.1%; Fe, by 54.0%; V, by 31.6%). The results provide empirical evidence for the application potential of fungal extracellular enzymes from *Aspergillus* spp. in EHCO recovery and biocatalytic upgrading of EHCO.

## 1. Introduction

Heavy and extra-heavy crude oils (HCOs and EHCOs, respectively) are the largest-known potentially recoverable petroleum resources, accounting for ~70% of global oil reserves [1]. Unconventional HCO and EHCO reserves have attracted increasing attention due to the growth in global energy demand and the depletion of conventional crude oil reserves. Efficient development of HCO and EHCO reservoirs is essential for easing the pressure of light crude oil shortage [2]. Cold production and thermal extraction are two classes of techniques used for enhanced recovery of HCO and EHC [3]. Cold production (e.g., chemical, biological, compound methods) is superior to thermal production (e.g., steam flooding, in situ combustion) in terms of oil extraction efficiency and energy consumption [4]. In recent decades, microbial-enhanced oil recovery (MEOR) has emerged as a cost-effective technique with successful applications in the exploitation of depleted oil fields [5,6].

MEOR involves the injection of microbes together with metabolites or nutrients into the oil reservoir to increase the swept volume, oil displacement efficiency, and consequently oil recovery [7]. This technique demonstrates the potential for heavy crude oil recovery by using bacteria (e.g., *Pseudomonas*, *Bacillus*, *Dietzia*, *Chelatococcus*) that degrade heavy oil and produce bioproducts [7,8,9]. These bacteria are able to reduce oil viscosity via degradation/transformation of heavy petroleum hydrocarbons (e.g., waxes, resins, even asphaltenes) into smaller fractions, in addition to demetallization of asphaltenes. Some bacteria produce surfactants, solvents, organic acids, and gasses, which can promote crude oil mobility by forming stable oil–water emulsions and reducing surface and interfacial tensions between water and oil [10]. However, the MEOR performance of known bacteria is often unsatisfactory for HCO and EHCO characterized by extraordinarily high viscosity and abundant high-molecular-weight (HMW) compounds. In particular, microbial degradation or biotransformation of EHCO is limited by its high asphaltene content [9,11].

A growing number of fungi (e.g., *Pestalotiopsis*, *Aspergillus*, *Mucor*, *Penicillium*) have been shown to metabolize hydrocarbons via distinctive enzymatic mechanisms [12,13]. These fungi synthesize and release extracellular enzymes with outstanding catalytic activity, which are involved in the following: (1) partial transformation of hydrocarbons; (2) complete degradation of hydrocarbons with an additional carbon source; and (3) independent utilization of hydrocarbons as a sole carbon source. For example, the fungal enzymes responsible for lignin degradation (e.g., laccases, manganese peroxidases, lignin peroxidases) have greater potential than bacterial inoculants in bioremediation of soils contaminated with long-chain petroleum hydrocarbons, especially polycyclic aromatics [14]. Additionally, invasive hyphal growth of filamentous fungi can expand their contact area with petroleum hydrocarbons, thereby facilitating biodegradation. Despite their advantages over bacteria, the fungi known as hydrocarbon metabolizers may not be directly applicable to anoxic reservoirs due to insufficient oxygen supply [15]. An alternative approach is to apply fungal extracellular enzymes from hydrocarbon-degrading taxa as biocatalysts for heavy oil degradation.

Successful use of enzymes for crude oil recovery has been reported in many countries and regions, including China, Indonesia, Myanmar, and Venezuela. In China’s Shengli oilfield, an enzyme-based agent composed of proteinases and bacteria was applied for plug removal in oil wells [16]. The applications of fungal enzymes mainly involve biocatalytic cracking—a technology used to catalyze biochemical reactions. As for the biodegradation of petroleum hydrocarbons, heme-chloroperoxidase from *Caldariomyces fumago* promotes oxidation of a petroporphyrin-rich fraction in asphaltenes in the presence of hydrogen peroxide, accompanied by efficient removal of nickel (Ni) and vanadium (V) [17]. Lignin peroxidase of the extremophilic fungus *Pestalotiopsis palmarum* BM-04 catalyzes oxidation of carbon and sulfur atoms in the maltene and asphaltene fractions of EHCO [18]. Extracellular enzymes from *Aspergillus* spp. reduce the viscosity of HCO by oil degradation with gas formation [19]. Given their outstanding catalytic activity, fungal extracellular enzymes, at least to a certain degree, provide a technically feasible approach to upgrade heavy oil. The use of fungal extracellular enzymes thus represents a novel concept for HCO and EHCO exploitation.

Previously, two hydrocarbon-degrading *Apergillus* strains, *A. terreus* HJ2 and *A. nidulans* HJ4, were isolated from bitumen. Their extracellular enzymes exhibited dehydrogenase and catechol 2,3-dioxygenase activities, and enzymatic degradation resulted in lower viscosity and redistributed fractions in HCO [19]. This begs the question of whether these fungal enzymes exhibit the same effects on EHCO, which is more viscous and less biodegradable than HCO. The aim of this study was to investigate the capacity and mechanism of fungal extracellular enzymes from *Aspergillus* spp. for EHCO recovery. We analyzed the biodegradation and biotransformation of hydrocarbons, formation of biogases and organic acids, reduction in viscosity, and efficiency of demetallization in EHCO treated with fungal enzymes from each strain separately and mixtures of enzymes. The application potential of fungal enzymes for EHCO recovery was also determined. Results of this study could provide guidance on the use of fungal extracellular enzymes as low-cost and ecologically friendly biocatalysts for EHCO recovery.

## 2. Materials and Methods

### 2.1. Fungal Strains and Oil Sample

We selected two bitumen-derived fungal strains, i.e., *A. terreus* HJ2 (sequence No.: MG732935) and *A. nidulans* HJ4 (sequence No.: MG732936). Both strains were maintained on potato dextrose agar (PDA) slants at 4 °C and were reactivated in the PDA medium without agar at 30 °C for 3 days before use. The PDA medium (pH 7.0) was formulated with an extract from 200 g L^−1^ potato and 20 g L^−1^ dextrose [20].

An EHCO sample was obtained from the No. 2 Oil Production Plant in Tahe Oilfield (Xinjiang, China). The four generic fractions (saturates, aromatics, resins, and asphaltenes, SARA) in the EHCO sample were 1.66 ± 0.43%, 35.51 ± 1.96%, 8.39 ± 2.26%, and 54.44 ± 0.80%, respectively, by weight. The oil viscosity was 200,457 mPa·s (50 °C).

### 2.2. Oil Biodegradation and Medium Optimization

To ascertain the ability of strains HJ2 and HJ4 to biodegrade EHCO, we measured their growth with EHCO as the sole carbon source. Briefly, mycelial pellets were collected from the PDA medium without agar and inoculated onto agar plates prepared with mineral salts solution (MSS) containing 100 g L^−1^ of EHCO. MSS (pH 7.0) was formulated with 3.00 g (NH_4_)_2_SO_4_, 0.50 g MgSO_4_·7H_2_O, 5.00 g NaCl, 2.00 g KH_2_PO_4_, and 3.00 g K_2_HPO_4_·3H_2_O per liter. All plates were incubated at 30 °C, with fungal growth observed and photographed between 7 and 14 days.

The medium for EHCO biodegradation was optimized by testing fungal growth with different carbon and nitrogen substrates (five each). Carbon source optimization was performed with MSS (pH 7.0) containing 100 g L^−1^ EHCO and 400 mg L^−1^ mannitol (or sucrose, glucose, soluble starch, *n*-hexadecane). Fungal growth diameters were measured at 30 °C over 14 days. The optimal carbon source was selected based on the maximum growth diameter, and the carbon concentration was adjusted from 100 to 500 mg L^−1^ to determine the optimal level. Nitrogen source optimization was performed with MSS containing 100 g L^−1^ EHCO and 400 mg L^−1^ optimal carbon source, with the original nitrogen source replaced by 3 g L^−1^ (NH_4_)_2_SO_4_ (or urea, NH_4_Cl, yeast extract, peptone). Fungal growth was measured over a 7-day period. Each test had three replications, and the results were averaged.

### 2.3. Enzymatic Degradation Test

Fungal enzyme preparations (mixtures of extracellular enzymes) were formulated by solid-state fermentation with the optimal carbon and nitrogen sources [19]. The fermentation medium (pH 7.0) was prepared with MSS (containing 0.4 g L^−1^ soluble starch and 3 g L^−1^ yeast extract), wheat bran, and EHCO (80:100:2, *v*/*w*/*w*). Two fungal strains (HJ2 and HJ4) were separately transferred to fresh PDA slants (4 mL) for activation, and the resulting spores in each tube were inoculated into the fermentation medium (50 g). The cultures were incubated at 28 °C for 5 days until white hyphae spread over the surface of the solid medium with a small number of spores. After that, the material containing fungal cultures and extracellular enzymes was cut into 1 cm pieces, followed by air-drying (~30.0% moisture) and oven-drying (40 °C, 48 h). The oven-dried material (~5.8% moisture) was grounded and sieved (0.10 mm).The powders from three replicate batches were combined as enzyme preparations, which were designated E2 (for HJ2) and E4 (for HJ4).

Crude enzyme extracts (2.5%, *w*/*v*) were prepared by dissolving the enzyme preparations in sterile distilled water and oscillating at 100 rpm and 30 °C for 15 h. The enzyme extracts were filtered through sterile 11 µm fiberglass membrane filters before use. An enzymatic degradation test was conducted in 250 mL glass bottles containing 2 g of EHCO and 40 mL of crude enzyme extract. The bottles were sealed with a rubber stopper and kept at 40 °C for 7 days, as a preliminary test showed gas formation mainly in the first 7 days with no major change in solution pH afterwards (Appendix A). To prevent the separation of an EHCO layer from the aqueous solution, all bottles were manually shaken once every day. At the end of the test, a needle attached to a 60 mL syringe (1 mL resolution) was inserted into the headspace of each bottle to measure the total gas yield. The residual oil was extracted with *n*-hexane to analyze SARA fractions as described previously [21].

### 2.4. Optimization of Enzymatic Degradation Conditions

#### 2.4.1. Enzyme Concentration Optimization

To optimize the enzyme concentration for EHCO degradation, we performed an enzymatic degradation test in 250 mL ground glass bottles containing 1 g of EHCO and 40 mL of crude enzyme extract at gradient concentrations (2.5–10.0%, *w*/*v*). Controls were prepared by replacing the crude enzyme extract with an equal volume of sterile distilled water. Biogas yield was recorded after 7 days of degradation to determine the optimal enzyme concentration.

#### 2.4.2. Enzymatic Complex Optimization

The enzyme preparations from multiple strains are expected to have greater effects than those from each strain separately [22]. Therefore, we prepared enzymatic complexes by mixing fungal enzymes from *Aspergillus* spp. with broad degradative effects. The composition of enzymatic complexes was adjusted by changing the volume ratio of E2 and E4 (Table 1). Briefly, 40 mL of mixed enzyme extracts (10.0%, *w*/*v*) were transferred into 250 mL ground glass bottles containing 2 g of EHCO for enzymatic degradation (7 days, 40 °C). The EHCO degradation capacity of different enzymatic complexes was assessed in terms of biogas yield.

#### 2.4.3. Optimized Enzymatic Degradation Test

To qualitatively evaluate the biodegradation capacity of enzymatic complexes, we performed the enzymatic degradation test in 250 mL ground glass bottles with 2.00 g of EHCO and 40 mL of crude enzyme extract (E2, E4, or E242; 2.5%, *w*/*v*). An equal volume of distilled water was used to replace the crude enzyme extract in controls. After enzymatic degradation (7 days, 40 °C), a 60 mL syringe was used to measure biogas yield and withdraw gas and liquid samples from each bottle.

The gas components were identified via gas chromatography (GC), using the method of Zhang et al. [23]. The analysis was accomplished on the Agilent 5977B series GC system (Agilent, Shanghai, China) equipped with an Agilent DB-5MS column (30 m × 0.25 mm × 0.25 µm) and a flame ionization detector. With regard to organic acid formation, we measured pH in liquid samples and identified the acid components via GC–MS with trimethylsilyl (TMS) derivatization. Specifically, 0.5 mL of liquid samples was lyophilized overnight in 1.5 mL deactivated vials. The dried solid samples were mixed with 0.5 mL of dimethyl sulfoxide to facilitate dissolution, and then added with 1 mL of bis-(trimethylsilyl)trifluoroacetamide with trimethylchlorosilane (99:1, *v*/*v*). The vials were capped and placed in a 65 °C water bath for 2 h to allow complete silylation. After that, the samples were injected into an Agilent 7890B series GC/MS system equipped with an Agilent DB-Petro column (50 m × 0.2 mm × 0.5 µm) and a time-of-flight mass spectrometer.

The residual oil was extracted from liquid samples with *n*-hexane, and its SARA fractions were analyzed using Al_2_O_3_ column chromatography. The oil components were identified by GC–MS using the methods of Zhang et al. [19]. The concentrations of heavy metals in EHCO samples were determined by inductively coupled plasma–optical emission spectroscopy (ICP–OES) based on the method of Suganthi et al. [24]. Viscosity measurements of EHCO were performed using the method described by Zhang et al. [19].

### 2.5. Data Analysis

Measurement data are presented as the mean ± standard deviation (*n* = 3). All data were statistically analyzed using SAS v9.2 (SAS Institute Inc, Cary, NC, USA). To test for the differences between group means, one-way analysis of variance with Duncan’s multiple range test was conducted at the *p* < 0.05 level. Origin v2022 (OriginLab Corp., Northampton, MA, USA) was used to create graphs.

## 3. Results and Discussion

### 3.1. Fungal Degradation of Extra-Heavy Oil

While increasing research attention has been drawn to heavy oil biodegradation, many studies focus on light crude oil, its low-molecular-weight (LMW) components, or purified fractions [20,25]. There is still a paucity of studies exploring the biodegradation of HCO and EHCO, especially by fungi [26]. In the present study, *A. terreus* HJ2 and *A. nidulans* HJ4 isolated from bitumen showed robust growth with EHCO as the only carbon and energy source. The ability of these two fungi to utilize EHCO was indicated by their large growth diameters, with an average of 74.5 mm (HJ2) and 75.5 mm (HJ4) at 14 days (Figure 1). This indicates that both *A. terreus* HJ2 and *A. nidulans* HJ4 are promising biodegraders of EHCO.

A handful of fungal strains isolated from extreme habitats can degrade asphaltenes and HMW polycyclic aromatic hydrocarbons. For example, *Pestalotiopsis palmarum* BM-04 isolated from an asphalt lake produces oxidative exoenzymes to catalyze the biodegradaton of maltenes, asphaltenes, and the petroporphyrin-rich fraction of EHCO [18]. Additionally, *Alternaria* sp. isolated from deep-sea sediment utilizes EHCO as the sole carbon source and effectively degrades aromatics in EHCO [26]. The ability of fungi to degrade and/or transform petroleum hydrocarbons must be taken into consideration when selecting strains for EHCO recovery [27]. Given their robust growth with EHCO as a sole carbon source, *A. terreus* HJ2 and *A. nidulans* HJ4 are potential candidates for EHCO biodegradation, biotransformation, and additional recovery.

### 3.2. Medium Optimization for Oil Biodegradation

HCO, particularly EHCO, that contains abundant resins and asphaltenes is recalcitrant to microbial degradation under natural conditions [28]. The addition of exogenous organic substrates or other nutrients as an auxiliary carbon and energy source has the potential to accelerate the biodegradation of recalcitrant heavy hydrocarbons [13]. To improve the biodegradation efficiency of EHCO by *A. terreus* HJ2 and *A. nidulans* HJ4, five carbon sources were tested as co-substrates. When supplemented with extra carbon sources, the fungal growth was promoted effectively based on increased growth diameters (Figure 1). This result is in agreement with a previous study showing that the dissociation of HCO by *Purpureocillium lilacinum* and *Penicillium chrysogenumwas* was accelerated in the presence of co-substrates [21].

When *A. terreus* HJ2 was grown on the medium supplemented with additional carbon sources, its mean growth diameter varied from 74.0 mm (mannitol) to 82.0 mm (soluble starch). The mean growth diameter of *A. nidulans* HJ4 was between 77.5 mm (glucose) and 83.5 mm (soluble starch; Table 2). The results indicate that soluble starch is the preferred auxiliary carbon source for stimulating the growth of both *A. terreus* HJ2 and *A. nidulans* HJ4 on EHCO agar plates. Next, the effect of soluble starch concentration on fungal growth was evaluated. Both fungal strains exhibited excellent growth performance on EHCO agar plates supplemented with 400 mg L^−1^ of soluble starch (Table 2). Therefore, soluble starch (0.4 g L^−1^) was selected as an external carbon source to enhance fungal growth on EHCO.

Furthermore, the optimal nitrogen source for the two fungal strains was screened (Table 2). *A. terreus* HJ2 showed the best growth on EHCO agar plates with organic nitrogen sources, where the mean growth diameters reached 59.0 mm (yeast extract) and 55.0 mm (peptone). The growth diameters of HJ2 with mineral nitrogen sources were smaller than 40 mm. In the case of *A. nidulans* HJ4, the best nitrogen sources were also yeast extract and peptone (diameter: 73.5 and 70.5 mm, respectively). The supply of (NH_4_)_2_SO_4_ and NH_4_Cl resulted in inferior growth of HJ4 (diameter: 48.5 and 48.0 mm, respectively), and the worst growth was observed with urea (diameter: 43.0 mm). The results imply that organic nitrogen, rather than mineral nitrogen, is preferential for both fungal strains (Figure 1), and yeast extract is the optimal nitrogen source for fungal growth on EHCO. Accordingly, we selected soluble starch and yeast extract for the two fungi in subsequent experiments.

### 3.3. Oil Degradation Capacity of Fungal Enzymes

It has been well documented that fungal enzymes have a great ability to degrade light crude oil and HCO [19,23]. To determine the degradation capacity of fungal enzymes for EHCO, we measured the changes in four SARA fractions and total gas yield after enzymatic treatment. Table 3 provides the efficiency of E2 and E4 to degrade EHCO. Both of the enzyme preparations from *Aspergillus* spp. exhibited EHCO degradation activity, as demonstrated by the redistribution of SARA fractions and simultaneous formation of biogases. Compared to the control group, the content of saturates markedly increased by 23.2–60.0% after E2 treatment, with a smaller increase in the content of aromatics (by 2.8%). In contrast, there was a 9.5% decrease in the content of aromatics after E4 treatment. In both the E2 and E4 treatment groups, the contents of resins and asphaltenes decreased by 32.3–53.9% and 5.8–20.8%, respectively.

According to Raheem et al. [29], an increase in saturate and aromatic fractions coupled to any changes in resin and asphaltene fractions would support the biotransformation of the biotreated crude oil. The changes we observed in the SARA fractions of EHCO suggest that some chemical groups were transferred among these four fractions due to enzymatic degradation. This redistribution of heavy hydrocarbons could enable their transformation into lighter ones, thus improving the physicochemical properties of the biotreated EHCO. Particularly, the decrease in resins and asphaltenes would lead to a reduction in oil viscosity and freezing point, increasing the oil mobility in situ [2]. The EHCO recovery technique involving enzymatic degradation of asphaltenes is of practical value and needs to be further explored.

Considerable yield of biogases was observed in the process of EHCO degradation with the two enzyme extracts at a 2.5% concentration. The total gas yield with E2 and E4 was 34.8 and 27.2 mL 100 mL^−1^, respectively (Table 3). GC analysis identified the major gas components as CO_2_ and H_2_. Kraemer and Bagley [30] reported that in the absence of oxygen, microbes are responsible for fermentative formation of CO_2_, H_2_, acids, and alcohols. As there were no live microbes in our reaction system, the formation of biogases, particularly H_2_, was most likely attributed to fungal dehydrogenase activity [19]. In addition to dehydrogenases, hydrogenases have been reported as important metalloenzymes for H_2_ evolution and CO_2_ reduction in nature [31,32]. Identifying the enzymes (or proteins) present in the crude enzyme extracts can give more information on the mechanisms of biogas formation during ECHO degradation. The biogases formed in situ are expected to increase the pressure in oil reservoirs and consequently facilitate oil extraction. Furthermore, these biogases may dissolve into the crude oil, and as such, reduce its dynamic viscosity [33]. Our results are encouraging for the use of fungal enzymes in exploitation and downstream processing of EHCO.

### 3.4. Optimization of Enzyme Concentration and Combination

To achieve higher efficiency and broader spectrum of EHCO degradation, we optimized the concentration and combination of fungal enzymes in terms of biogas yield. The total gas yield after 7 days of EHCO degradation at different enzyme concentrations is shown in Figure 2. As the enzyme concentration increased from 2.5% to 10.0%, the total gas yield trended upward from 48.4 to 129.0 mL 100 mL^−1^ (E2) and from 41.6 to 126.4 mL 100 mL^−1^ (E4). The results imply that within the range tested, a higher enzyme concentration supports a higher yield of biogases from EHCO. Based on biogas yield, we selected 10.0% as the optimal enzyme concentration for EHCO degradation.

The biogas yields from EHCO with different enzymatic complexes are provided in Table 1. Compared to E2 and E4 only, their combinations at different ratios generally improved the total gas yield (except for E244). When treated with enzymatic complexes, the total gas yield reached a maximum of 132.0 mL 100 mL^−1^ (E242). Increased yield of biogases can contribute to improved oil recovery by reducing oil viscosity and re-pressurizing the reservoir [34]. Therefore, E242 comprised of E2 and E4 (5:3, *v*/*v*) was selected as the optimal enzymatic complex for EHCO degradation.

### 3.5. Optimized Enzymatic Degradation of Extra-Heavy Oil

#### 3.5.1. Distribution of SARA Fractions

EHCO can be classified into four SARA fractions using Al_2_O_3_ column chromatography. The efficiency of E2, E4, and E242 to degrade the four SARA fractions is provided in Table 4. In the E2 and E4 treatments, the content of saturates increased by 25.0% and 43.8%, respectively, compared to that of the control group. A similar upward trend was observed for aromatics (by 36.9% and 29.4%, respectively). Conversely, the content of resins decreased by 52.4% and 55.7% in the E2 and E4 treatments, respectively, corresponding to the downward trend in asphaltenes (by 33.5% and 42.8%, respectively). In summary, the treatment with fungal enzymes from each strain separately resulted in a redistribution of SARA fractions from resins and asphaltenes to saturates and aromatics in EHCO. In the E242 treatment, the content of saturates decreased by 18.8%, in contrast to the results from the E2 or E4 treatment. The variation patterns in the other three fractions (i.e., aromatics, resins, asphaltenes) were comparable to those observed with E2 or E4. Specifically, the contents of resins and asphaltenes decreased by 61.8% and 48.0%, respectively, whereas the content of aromatics increased by 45.1%. E242 achieved higher degradation rates of resins and asphaltenes than E2 and E4 when applied in the same quantities. This implies that the enzymatic complex has a greater ability to degrade EHCO compared to fungal enzymes from each strain separately.

Very few microbes are known to degrade asphaltenes, which are the most recalcitrant crude oil fraction. Only a handful of studies have reported on the biodegradation or mineralization of asphaltenes by mixed bacterial consortia (e.g., [28]). Xia et al. [35] found that the relative contents of saturates, aromatics, resins, and asphaltenes in HCO were, respectively, reduced by 10.6%, 6.0%, and 3.6% after treatment with indigenous bacterial consortia composed of *Firmicutes*, *Proteobacteria*, *Deferribacteres*, and *Bacteroidetes*. Sun et al. [36] screened exogenous bacteria and combined them with indigenous bacteria to treat HCO, which resulted in decreased wax and resin–asphaltene contents (by 12.3% and 16.9%, respectively). Notably, the degradation/transformation rates of resins and asphaltenes by the enzymatic complex E242 (61.8% and 48.0%, respectively) are amongst the higher values reported previously [24,28,35]. Taken together, the results of SARA analysis indicate that the fungal enzymes from *A. terreus* HJ2 and *A. nidulans* HJ4 act as biocatalysts to transform polar HMW resins and asphaltenes into smaller, less polar, or nonpolar fragments in ECHO. This redistribution of heavy hydrocarbons could lead to viscosity reduction and fluidity improvement in crude oil, allowing for enhanced oil recovery.

#### 3.5.2. Composition of Gasifiable *n*-Alkanes

The quantitative changes in gasifiable *n*-alkanes in EHCO after enzymatic treatment with E2, E4, and E242 were revealed by the GC–MS analysis. Enzymatic degradation significantly altered the relative quantities of *n*-alkanes (Table 5). There were increased short- and medium-chain *n*-alkanes (C_8_–C_10_, C_18_–C_20_), accompanied by decreased long-chain *n*-alkanes (C_23_–C_26_, C_28_–C_30_), in the enzymatic treatments compared to the control group. In all cases, E2, E4, and E242 preferentially degraded long-chain *n*-alkanes (C_28_–C_30_), with the degradation rates of nonacosane and triacontanae being higher than 25.0%. These results provide solid evidence for enzyme-mediated changes in the distribution of *n*-alkanes from heavy fractions to light fractions, which could diminish oil viscosity.

While bacterial degradation of long-chain *n*-alkanes in HCO has been recently demonstrated [37], we, for the first time, report the biodegradation of long-chain *n*-alkanes in EHCO by fungal enzymes. Since our fungal strains were isolated from bitumen, an extremely sticky and highly viscous form of petroleum, they are expected to carry functional genes for preferential degradation of long-chain *n*-alkanes. The preference of fungal enzymes for long-chain *n*-alkanes has been reported in previous studies that focused on bioremediation of polluted soils [14,38]. For example, Lladó et al. [38] showed that *Trametes versicolor* is able to degrade heavy mineral oil (C_15_–C_35_ hydrocarbons). Furthermore, fungal enzymes from each strain separately (e.g., E2, E4) and a mixture of enzymes (e.g., E242) are capable of degrading a broad range of *n*-alkanes between C_8_ to C_30_ within a short duration (7 days), in contrast to conventional biodegradation tests using fungal or bacterial cultures [26]. The remarkable compositional changes in *n*-alkanes (C_8_–C_30_) and the short duration of enzymatic degradation demonstrate the superior performance of fungal enzymes in degrading EHCO.

The metabolic pathways of *n*-alkanes under aerobic conditions are well understood. The initial step involves oxidation with an oxygen atom inside the terminal methyl group to form a primary alcohol. The alcohol is then oxidized to aldehyde and subsequently to fatty acid. Further, the fatty acid is broken down into CO_2_, H_2_O, non-toxic, or less toxic residues via β-oxidation [14]. After oxygen depletion by aerobic degradation, anaerobes can transform organic compounds into useful products (e.g., CH_4_, acetic acid) using nitrate (NO_3_^−^), sulfate (SO_4_^2−^), and carbonate (CO_3_^2−^) as alternative electron acceptors [39]. Based on the known metabolic pathways of *n*-alkanes and the degradation results of EHCO (Table 4 and Table 5), we propose the possible mechanisms underlying *n*-alkane biodegradation by fungal enzymes in the following three steps.

First, resins and asphaltenes are enzymatically converted to more readily available fractions (e.g., saturates, aromatics), as has been reported for the biotransformation of crude oil by microbial consortia [29]. Second, long-chain *n*-alkanes (C_28_–C_30_) are enzymolyzed to medium-chain fractions (C_18_–C_20_) or directly to short-chain fractions (C_8_–C_10_), releasing large amounts of biogases (e.g., CO_2_, H_2_). Third, the medium- and short-chain fractions are degraded, with LMW acids (e.g., acetic acid, propionic acid) and biogases (e.g., CO_2_, H_2_) formed under anaerobic conditions, as detected by GC–MS. In MEOR, indigenous microbes or microbial products can convert long-chain *n*-alkanes to lighter alkanes, resulting in lower viscosity and improved flow properties of crude oil [34]. Identifying the metabolic mechanisms that underlie enzymatic degradation of long-chain *n*-alkanes in crude oil can facilitate the development of MEOR strategies for EHCO exploitation.

#### 3.5.3. Formation of Biogases and Organic Acids

Gas infusion has wide applications in the exploitation of light oils. Recently, gas injection has also emerged as a promising technique to enhance HCO recovery, where different methods such as natural gas injection and CO_2_ injection have been employed to improve the oil displacement efficiency [40]. Large amounts of biogases, mainly CO_2_ and H_2_, were produced during the enzymatic degradation of EHCO, as revealed by GC analysis. Under the optimized conditions, the total gas yield was 113.6 mL 100 mL^−1^ (E2), 109.2 mL 100 mL^−1^ (E4), and 135.6 mL 100 mL^−1^ (E242; Table 4). Compared to E2 and E4, the enzymatic complex E242 exhibited a higher ability to degrade EHCO with formation of more biogases, which could offer greater benefits to EHCO recovery. Biogases, such as CH_4_, CO_2_ and H_2_, can also be formed when microbes ferment petroleum hydrocarbons. These biogases serve to promote oil recovery from dead space and dislodge debris from pores through reduction in oil viscosity and re-pressurization of the reservoir [34,35]. For example, some bacteria (e.g., *Halanaerobium* sp., *Methanothermobacter* sp., *Bacillus* sp.) are found to ferment crude oil or pure hydrocarbons with the release of CO_2_, H_2_, and CH_4_. This can promote pressure buildup and create oil swelling, decrease the interfacial tension between two interfacing liquids, and consequently enhance sweep efficiency [33,35].

Gas injection, particularly CO_2_-enhanced oil recovery, has been successful for four decades, yet it still has production potential [1]. The fungal enzymes from each *Aspergillus* strain separately, as well as a mixture of enzymes, carried out the transformation of EHCO accompanied by substantial formation of biogases (Table 4). This agrees with a previous study that showed HCO degradation by the same fungal enzymes coupled with substantial gas formation [19]. The total gas yield during HCO and EHCO degradation by E2 and E4 is higher than previously reported for bacteria [35,41]. Under anaerobic conditions, 33.9 mL of CH_4_ was formed when enriched indigenous bacterial consortia degraded 2.34 g of HCO over a 200-day period [35]. Additionally, 103.0 mL of CH_4_ and 12.3 mL of CO_2_ were formed after 1.50 g of HCO was degraded by a consortium of indigenous archaea over a 250-day period [41]. A total of 335.3 mL of gasses (mainly CO_2_ and H_2_) were formed by degrading 2.00 g of EHCO with E242 in a 7-day period, unlocking the potential of this enzymatic complex for application in EHCO recovery.

Acids, primarily LMW fatty acids, are also important products or intermediate metabolites of microbial fermentation which can increase the porosity and permeability of oil reservoirs by dissolving carbonate rocks [34]. EHCO degradation by fungal enzymes was associated with substantial formation of acids, as evidenced by a notable decrease in the solution’s pH compared to the control group. The pH of the reaction solutions was 7.3 in the control samples, much higher than 5.7 in E2 treatment, 5.9 in E4 treatment, and 5.5 in E242 treatment (Table 4). Pereira et al. [42] observed a decrease in pH during hydrocarbon biodegradation, which was attributed to the formation of organic acids. Al-Hawash et al. [20] also showed that *n*-hexadecane degradation by *Aspergillus* sp. RFC-1 led to a decrease in the medium pH as a result of fatty acid formation. This was verified by the detection of 9-octadecadienoic acid (C_18_H_34_O_2_), heptadecanoic acid (C_17_H_34_O_2_), and hexadecanoic acid (C_16_H_32_O_2_), as well as other LMW fatty acids, including octanoic acid (C_8_H_16_O_2_) and glutaric acid (C_5_H_8_O_4_) in the *n*-hexadecane medium inoculated with strain RFC-1. These findings allow us to posit that organic acids were formed due to the enzymatic degradation of EHCO, which in turn affected the pH of the reaction solutions.

TMS derivatization analysis indicated that the organic acids formed in the biotreated EHCO were mainly monocarboxylic acids (e.g., acetic acid, propionic acid, butyric acid) and dicarboxylic acids (e.g., oxalic acid, malonic acid; Appendix A). These LMW organic acids are among useful microbial products that can facilitate MEOR. They can dissolve carbonate rocks to increase the reservoir permeability and porosity, allowing for access to hidden residual oils [34]. In contrast, some bacterial strains have little or no effect on the pH level during the fermentation process. Pereira et al. [42] observed a slight, but not significant, decrease in the medium pH during biodegradation of aliphatic hydrocarbons and polycyclic aromatic hydrocarbons by *B. methylotrophicus* and *P. sihuiensis*. Haddad et al. [5] found minimal changes in the solution pH during incubation and after treatment of crude oil with *B. persicus*. Together, the findings underscore the advantages of fungal enzymes from *Aspergillus* spp. for enhanced oil recovery, as they can promote biogas and organic acid formation when breaking down EHCO. Trapped oil can be liberated by microbial metabolites: gasses that increase pressure to force more oil out of wells and organic acids that dissolve carbonaceous deposits to increase the well permeability. However, Hillman et al. [43] and Madirisha et al. [44] proposed that microbial activity in oil wells may also be disruptive to oil recovery and/or alter oil composition. For example, microbes can degrade and sour oil through specific metabolites, such as H_2_S, formic acid, acetic acid, and methyl alcohol. These microbial metabolites are likely to corrode the well casing, flowlines, and pipelines, despite rarely reaching particularly low pH values. The potential threat causing the formation of undesirable metabolites needs to be carefully addressed and investigated.

#### 3.5.4. Demetallization of Extra-Heavy Oil

The majority of heavy metals in HCO and EHCO, particularly Ni, V, and iron (Fe), are present in the asphaltene fraction [45]. These metals generally exist in asphaltenes in two forms; metal porphyrins (C_27_N_4_–C_33_N_4_) are of biological origin, whereas nonporphyrins remain poorly understood [46]. Biodegradation of asphaltenes by enzymes renders the metals accessible for biocatalytic reaction [45]. Treatment with the fungal enzymes from each *Aspergillus* strain separately and a mixture of enzymes caused a decrease in the concentrations of heavy metals in EHCO (Table 6). ICP-OES results showed that the initial concentrations of Ni, Fe, and V in EHCO were 12.7, 136.5, and 227.5 mg kg^−1^, respectively. After treatment with E2, E4, and E242, the removal rate of Ni reached 9.4%, 13.4%, and 44.1%, respectively; the removal rate of Fe reached 45.2%, 27.1%, and 54.0%, respectively; the removal rate of V reached 20.4%, 10.5%, and 31.6%, respectively. The results provide robust evidence that the fungal enzymes are capable of removing heavy metals from EHCO, with the highest efficiency observed for the enzymatic complex E242. In this regard, enzymatic treatment enables simultaneous biodegradation and demetallization of EHCO. Further research should measure metal concentrations in the reaction solution to identify the mechanism of enzymatic demetallization for EHCO.

Undesirable oil properties, such as high viscosity and the propensity to form emulsions, are largely due to the asphaltenic fraction [9]. One strategy to improve the viscosity of HCO and enhance its mobility through the reservoir is breaking down and demetalizing asphaltene aggregates [47]. Despite some microbes being found to degrade asphaltenes, it is still challenging to remove metals from porphyrins trapped in the complex asphalteneic structure. While the degradation of metalloporphyrines has been related to microbes, no direct evidence is available for the biodemetalization of crude oil, and those compounds are recalcitrant to microbial attack [48]. Nzila and Musa [49] indicated that microbes may degrade the aliphatic and aromatic moieties of asphaltene compounds, leaving the most complex parts as intermediates; thus, biomineralization of asphaltenes still remains unresolved. Nevertheless, other researchers argued that some enzymes secreted by fungi, such as the cytochrome P450 and lignin-degrading enzymes, are able to locate on the extracellular membrane and launch a preliminary attack to asphaltenes at the cell surface, destroying the porphyrin structure and releasing V and Ni [18,50]. In this regard, the fungal enzymes from *Aspergillus* spp. are advantageous in that they can work together to remove heavy metals (Ni, Fe, V) in EHCO. The use of such enzymes (crude extracts or solid powders) will provide a new opportunity for heavy hydrocarbon degradation and demetallization of EHCO.

#### 3.5.5. Enzymatic Reduction in Oil Viscosity

Changing the oil properties, such as reducing the oil viscosity, is one of the important mechanisms in MEOR [8]. Conventional methods for viscosity reduction include heating, emulsification, and dilution with light crude oil or chemical diluents. Their application often requires high costs and causes potential damage to the reservoir formation [51]. Therefore, we examined the potential of fungal enzymes from each *Aspergillus* strain separately and a mixture of enzymes to reduce oil viscosity through biodegradation. All fungal enzymes applied at a final concentration of 10.0% (*w*/*v*) significantly reduced the viscosity of EHCO. The enzymatic complex E242 achieved greater viscosity reduction efficiency (12.7%) than E2 (5.1%) and E4 (4.5%; Table 4). E2 and E4 (4.0%, *w*/*v*/*v*) were also found to cause >50% reduction in the viscosity of HCO [19]. Moreover, some bacterial strains (e.g., *B. subtilis*, *B. licheniformis*, *Chelatococcus daeguensis*) were shown to reduce HCO viscosity by 15.0–59.8% under aerobic and aerobic conditions [35,36]. While many studies conducted biodegradation tests with HCO (API specific gravity >10, viscosity <30,000 mPa·s), the biodegradation of EHCO (API specific gravity <10, viscosity <210,000 mPa·s) is much more difficult to conduct. Compared to those HCO-degrading enzymes and viscosity-reducing bacteria, the fungal enzymes tested in the present study have superiority in dealing with EHCO. In particular, the enzymatic complex E242 can efficiently reduce the viscosity of EHCO.

Biocatalytic viscosity reduction in crude oil can be attributed to two possible mechanisms—the degradation of heavy fractions and the formation of bioproducts [5]. The fungal enzymes from *Aspergillus* spp., especially the enzymatic complex E242, exhibited prominent abilities to degrade long-chain *n*-alkanes, resins, and asphaltenes with the formation of CO_2_ and H_2_, as well as monocarboxylic and dicarboxylic acids, all of which contributed to the viscosity reduction in EHCO. Xia et al. [35] showed that the viscosity of dehydrated dead oil was reduced from 1823.86 to 1347.75 mPa·s as a consequence of heavy oil degradation and CH_4_ dissolution. Ke et al. [8] showed that *C. daeguensis* HB-4 effectively degraded long-chain hydrocarbons to shorter ones, resulting in lower oil viscosity. Despite bacterial cultures often being applied for crude oil degradation or MEOR, the processing period is long and the oil recovery efficiency is sometimes unsatisfactory, mainly because of poor bacterial growth in crude oil [10]. The fungal enzymes from *Aspergillus* spp. show excellent performance in cracking EHCO over a short period (7 days) under mild reaction conditions (e.g., neutral pH, moderate temperature), with no formation of hazardous waste. Moreover, there is no need to highly purify the fungal enzymes used EHCO biodegradation, as they can work together to implement multiple catalytic functions. Another advantage of enzymatic treatment is that the fungal enzymes are biodegradable proteins, which are unrecoverable after application and will eventually degrade in the environment. Therefore, the use of fungal enzymes from hydrocarbon-degrading strains is a feasible and promising approach to enhance EHCO recovery.

## 4. Conclusions

Two fungi isolated from bitumen, *A. terreus* and *A. nidulans*, exhibited vigorous growth with EHCO as a sole carbon source. Extracellular enzymes from each strain separately and a mixture of enzymes showed effective ability of degrading and demetallizating EHCO, as evidenced by decreased concentrations of heavy fractions (resins, asphaltenes) and heavy metals (Ni, Fe, V). The biodegradation process led to a decrease in oil viscosity with considerable formation of biogases and organic acids. The results indicate that fungal enzymes, particularly an enzymatic complex from two hydrocarbon-degrading strains, are promising candidates for EHCO degradation. This study presents a technically feasible and desirable procedure for the biotransformation of heavy hydrocarbons into lighter fractions using fungal enzymes.

Findings of this study imply that fungal enzymes from biodegradative strains can be used for exploitation and downstream processing of EHCO. Characterization of enzyme stability under different conditions (e.g., initial pH, temperature, incubation time) is crucial for enzyme preservation and application. The potential of fungal enzymes in field applications still needs to be verified by oil recovery tests in sand-pack columns and actual reservoir cores. Simulated-distillation analysis can be used to unveil the impacts of biotransformations on the refining operations.

## Figures and Tables

**Figure 1 microorganisms-12-02248-f001:**
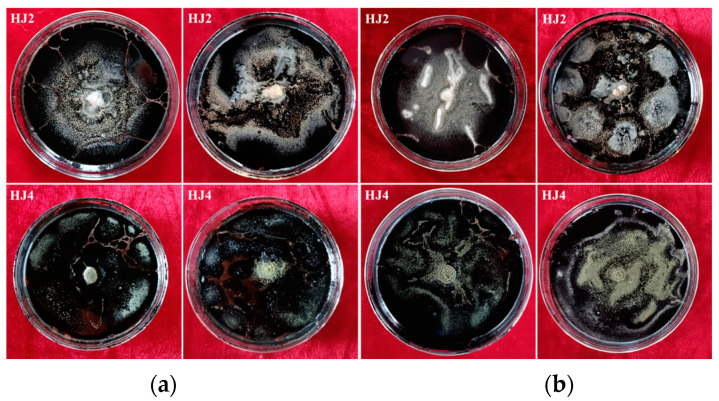
Growth of *A. terreus* HJ2 and *A. nidulans* HJ4 on the mineral basal medium (**a**) containing extra-heavy crude oil as a sole carbon source (left, 14 days), with 0.2% soluble starch as an additional carbon source (right, 14 days) and (**b**) containing extra-heavy oil and soluble starch, with (NH_4_)_2_SO_4_ (left, 7 days) and yeast extract (right, 7 days) as nitrogen sources.

**Figure 2 microorganisms-12-02248-f002:**
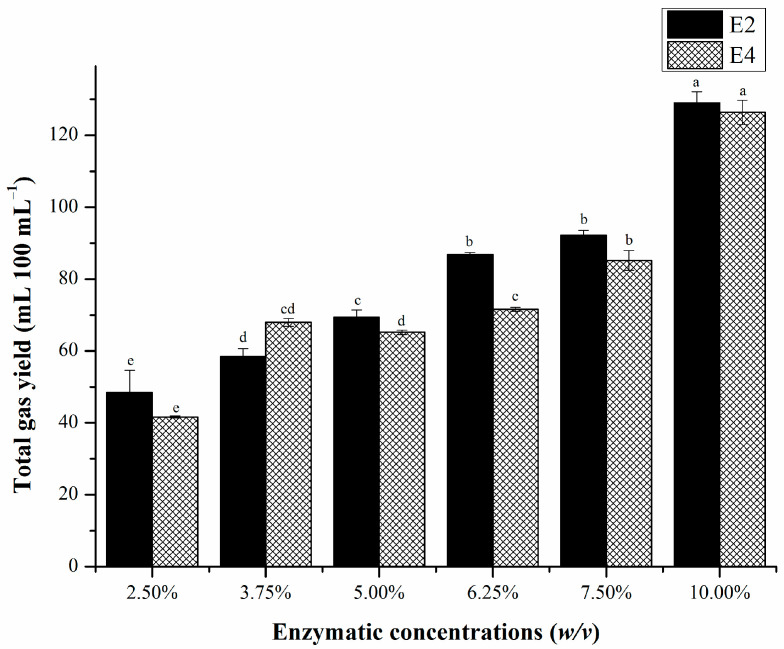
Total gas yield (mL 100 mL^−1^) from extra-heavy oil within 7 days of enzymatic treatment at different concentrations. Error bars represent standard deviation of the mean (*n* = 3). Different lowercase letters above the error bars indicate significant differences between groups (*p* < 0.05) determined by Duncan’s multiple-range test.

**Table 1 microorganisms-12-02248-t001:** Experimental design for optimization of fungal enzymatic complexes based on total gas yield.

Enzymatic Complex	E2 (mL)	E4 (mL)	Total Gas Yield (mL 100 mL^−1^)
E2	40	0	115.0 ± 1.4 ^cd^
E4	0	40	112.6 ± 0.8 ^d^
E241	30	10	123.8 ± 3.1 ^b^
E242	25	15	132.0 ± 1.1 ^a^
E243	20	20	122.2 ± 0.3 ^b^
E244	15	25	115.0 ± 1.4 ^cd^
E245	10	30	117.8 ± 0.3 ^c^

E2 and E4 are fungal enzymes from *A. terreus* HJ2 and *A. nidulans* HJ4, respectively. E241 to E245 are mixtures of E2 and E4 with different ratios. Values are presented as means ± standard deviation (*n* = 3). Different superscript letters within the same column indicate significant differences between groups (*p* < 0.05) determined by Duncan’s multiple-range test.

**Table 2 microorganisms-12-02248-t002:** Growth diameters (mm) of *A. terreus* HJ2 and *A. nidulans* HJ4 grown on the extra-heavy oil agar plates supplemented with different carbon or nitrogen sources and with different carbon concentrations for 7–14 days.

Fungus	Control	Mannitol	Sucrose	Glucose	Soluble Starch	Hexadecane
HJ2	74.5 ± 0.7 ^bc^	74.0 ± 0.0 ^c^	77.5 ± 0.7 ^bc^	76.0 ± 1.4 ^bc^	82.0 ± 2.8 ^a^	78.5 ± 2.1 ^ab^
HJ4	75.5 ± 0.7 ^c^	78.5 ± 2.1 ^b^	78.5 ± 0.7 ^b^	77.5 ± 0.7 ^bc^	83.5 ± 0.7 ^a^	77.8 ± 1.1 ^bc^
Fungus	Urea	NH_4_Cl	(NH_4_)_2_SO_4_		Yeast Extract	Peptone
HJ2	32.5 ± 0.7 ^b^	33.5 ± 2.1 ^b^	37.5 ± 3.5 ^b^		59.0 ± 2.8 ^a^	55.0 ± 1.4 ^a^
HJ4	43.0 ± 1.4 ^c^	48 ± 2.8 ^bc^	48.5 ± 0.7 ^b^		73.5 ± 2.1 ^a^	70.5 ± 2.1 ^a^
Fungus	Soluble Starch Concentration (mg L^−1^)					
	100	200	300		400	500
HJ2	44.5 ± 0.7 ^b^	47.0 ± 0.0 ^b^	44.5 ± 2.1 ^b^		54.0 ± 2.8 ^a^	42.0 ± 2.8 ^b^
HJ4	67.0 ± 4.2 ^b^	63.5 ± 3.5 ^b^	75.5 ± 0.7 ^a^		78.0 ± 0.0 ^a^	76.0 ± 1.4 ^a^

Values are presented as means ± standard deviation (*n* = 3). Different superscript letters within the same row indicate significant differences between groups (*p* < 0.05) determined by Duncan’s multiple range test.

**Table 3 microorganisms-12-02248-t003:** Four fractions of extra-heavy oil and gas yield after treatment with fungal enzyme extracts from *Aspergillus* spp.

Enzyme	Saturates		Aromatics		Resins		Asphaltenes		Gas Yield (mL 100mL^−1^)
	Content (g kg^−1^)	*VR*%	Content (g kg^−1^)	*VR*%	Content (g kg^−1^)	*VR_m_%*	Content (g kg^−1^)	*VR*%	
Ctrl	25.0 ± 3.0 ^b^	–	377.0 ± 9.8 ^a^	–	31.0 ± 11.5 ^a^	–	276.3 ± 6.4 ^a^	–	0.0 ± 0.0 ^c^
E2	30.8 ± 3.6 ^ab^	23.2	387.7 ± 4.0 ^a^	2.8	14.3 ± 4.0 ^a^	−53.9	260.3 ± 4.5 ^b^	−5.8	34.8 ± 0.6 ^a^
E4	40.0 ± 8.7 ^a^	60.0	341.3 ± 4.2 ^b^	−9.5	21.0 ± 7.9 ^a^	−32.3	218.7 ± 9.3 ^c^	−20.8	27.2 ± 0.6 ^b^

Values are presented as means ± standard deviation (*n* = 3). Different superscript letters within the same row indicate significant differences between groups (*p* < 0.05) determined by Duncan’s multiple range test. *VR*% represents enzyme-mediated variation (increase or decrease) in the mass of crude oil fractions.

**Table 4 microorganisms-12-02248-t004:** Enzymatic degradation of extra-heavy oil coupled to biogas and organic acid formation under optimized conditions.

Enzyme	Saturates		Aromatics		Resins		Asphaltenes		Gas Yield (mL 100 mL^−1^)	pH	Viscosity (mPa·s, 50 °C)
	Content (g kg^−1^)	*VR*%	Content (g kg^−1^)	*VR*%	Content (g kg^−1^)	*VR*%	Content (g kg^−1^)	*VR*%			
Ctrl	24.0 ± 2.8 ^bc^	–	414.5 ± 14.8 ^c^	–	36.1 ± 1.1 ^a^	–	525.5 ± 14.8 ^a^	–	0.0 ± 0.0 ^c^	7.3 ± 0.2 ^a^	219,917 ± 265 ^a^
E2	30.0 ± 4.2 ^ab^	25.0	567.5 ± 17.7 ^ab^	36.9	17.2 ± 0.3 ^b^	−52.4	349.5 ± 3.5 ^b^	−33.5	113.6 ± 5.6 ^b^	5.7 ± 0.2 ^bc^	208,634 ± 310 ^c^
E4	34.5 ± 3.5 ^a^	43.8	536.5 ± 13.4 ^b^	29.4	16.0 ± 1.3 ^b^	−55.7	300.5 ± 13.4 ^c^	−42.8	109.2 ± 2.8 ^b^	5.9 ± 0.2 ^b^	209,916 ± 246 ^b^
E242	19.5 ± 0.7 ^c^	−18.8	601.5 ± 16.3 ^a^	45.1	13.8 ± 2.1 ^b^	−61.8	273.5 ± 4.9 ^c^	−48.0	135.6 ± 0.6 ^a^	5.5 ± 0.1 ^c^	191,982 ± 276 ^c^

Values are presented as means ± standard deviation (*n* = 3). Different superscript letters within the same row indicate significant differences between groups (*p* < 0.05) determined by Duncan’s multiple range test. *VR*% represents enzyme-mediated variation (increase or decrease) in the mass of crude oil fractions.

**Table 5 microorganisms-12-02248-t005:** Redistribution of *n*-alkane fractions in extra-heavy oil after enzymatic treatment under optimized conditions.

10	Molecular Formula	Peak Area Control	Peak Area E2	*VR*%	Peak Area E4	*VR*%	Peak Area E242	*VR*%
666.9	C_8_H_18_	20521877	60677294	195.7	70622559	244.1	31914129	55.5
930.9	C_9_H_20_	41639690	83232858	99.9	89184939	114.2	50432318	21.1
1228.2	C_10_H_22_	77025332	96604881	25.4	96771714	25.6	67523231	−12.3
1532.3	C_11_H_24_	124826154	118741107	−4.9	118016663	−5.5	94388129	−24.4
1828.6	C_12_H_26_	149939343	142988803	−4.6	127503319	−15.0	119400393	−20.4
2110.9	C_13_H_28_	81834307	162874095	99.0	134310401	64.1	131344208	60.5
2378.1	C_14_H_30_	91025810	96708274	6.2	74494546	−18.2	138187801	51.8
2630.6	C_15_H_32_	93209279	109261802	17.2	82077634	−11.9	83442762	−10.5
2869.1	C_16_H_34_	92807001	113065316	21.8	148850547	60.4	85016396	−8.4
3094.9	C_17_H_36_	95303442	110495600	15.9	81427802	−14.6	82632888	−13.3
3309.1	C_18_H_38_	81685600	93869710	14.9	65491676	−19.8	118353162	44.9
3512.6	C_19_H_40_	72847916	95897799	31.6	102272933	40.4	105502871	44.8
3706.5	C_20_H_42_	62070413	73961924	19.2	51194116	−17.5	91494234	47.4
3891.4	C_21_H_44_	49598459	58845946	18.6	41151854	−17.0	41616559	−16.1
4068.0	C_22_H_46_	40782024	49018759	20.2	32948861	−19.2	59807620	46.7
4237.1	C_23_H_48_	32319673	41696313	29.0	27561027	−14.7	28983448	−10.3
4399.1	C_24_H_50_	27306509	32165207	17.8	20882672	−23.5	23077775	−15.5
4554.8	C_25_H_52_	21183514	24235973	14.4	17221248	−18.7	26780442	26.4
4704.2	C_26_H_54_	16020138	21817694	36.2	15222046	−5.0	15653326	−2.3
4848.3	C_27_H_56_	15112181	14260089	−5.6	11804840	−21.9	21566349	42.7
4986.9	C_28_H_58_	10965407	10685022	−2.6	9715957	−11.4	8847915	19.3
5072.4	C_29_H_60_	5056415	4501824	−11.0	3471125	−31.4	3447995	−31.8
5122.7	C_30_H_62_	8483866	6236466	−26.5	6293331	−25.8	6368784	−24.9

*VR*% represents enzyme-mediated variation (increase or decrease) in the relative quantity (peak area) of *n*-alkane fractions.

**Table 6 microorganisms-12-02248-t006:** Heavy-metal concentrations in extra-heavy oil after enzymatic treatment.

Enzyme	Ni		Fe		V	
	Concentration (mg kg^−1^)	Removal Rate (%)	Concentration (mg kg^−1^)	Removal Rate (%)	Concentration (mg kg^−1^)	Removal Rate (%)
Ctrl	12.7 ± 0.6 ^a^	–	136.5 ± 19.2 ^a^	–	227.5 ± 0.7 ^a^	–
E2	11.5 ± 0.4 ^a^	−9.4	74.8 ± 6.5 ^bc^	−45.2	181.0 ± 1.4 ^bc^	−20.4
E4	11.0 ± 1.4 ^ab^	−13.4	99.5 ± 11.3 ^b^	−27.1	203.5 ± 17.7 ^ab^	−10.5
E242	7.1 ± 2.5 ^b^	−44.1	62.8 ± 5.1 ^c^	−54.0	155.5 ± 6.4 ^c^	−31.6

Values are presented as means ± standard deviation (*n* = 3). Different superscript letters within the same row indicate significant differences between groups (*p* < 0.05) determined by Duncan’s multiple range test.

## Data Availability

The original contributions presented in the study are included in the article/Appendix A.

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
