# Peer review of "Fungal Extracellular Enzymes from *Aspergillus* spp. as Promising Candidates for Extra-Heavy Oil Degradation and Enhanced Oil Recovery"

_microorganisms, 2024, doi:10.3390/microorganisms12112248_

Round 1
Reviewer 1 Report
Comments and Suggestions for Authors
Some suggestions and objections related to the manuscript titled "Fungal Extracellular Enzymes from Aspergillus spp. as Promising Candidates for Extra-Heavy Oil Degradation and Enhanced Oil Recovery" are given below in the form of questions.
Please, describe solid-state fermentation procedure in the materials and methods section.
Section 3.5.2. Author discussed results of gasifiable n-alkanes composition and propose the possible mechanisms underlying n-alkaens biodegradation by fungal enzymes in the three steps. Are the proposed mechanism based on the results of biogas composition analyses or it is just assumptions? If the results of biogas composition exist, please include results in the manuscript.
Section 3.5.4. Please, indicate references with evidences of fungal enzyme demetallization. Mycoremediation is process used for the heavy metals depilation. How fungal enzymes depilate heavy metals in relative significant amount?
If authors measured, composition of fungal enzymes or at least concentration of proteins in the prepared fungal enzymes mixture will be welcome to the manuscript results. Potential enzymes for the analyses (activity determination) are: lacasses, MnP...
Author Response
Reviewer #1
Comments and Suggestions for Authors
Some suggestions and objections related to the manuscript titled "Fungal Extracellular Enzymes from Aspergillus spp. as Promising Candidates for Extra-Heavy Oil Degradation and Enhanced Oil Recovery" are given below in the form of questions.
Response: Thank you very much for kindly providing constructive comments on our manuscript. We have revised the text to address your concerns and additional changes are made as suggested by another reviewer. Please see our point-by-point response for details.
Comment 1)
Please, describe solid-state fermentation procedure in the materials and methods section.
Response: We have described the solid-state fermentation procedure in the materials and methods section as suggested.
Lines 145-152, Added, “Two fungal strains (HJ2 and HJ4) were separately transferred to fresh PDA slants (4 mL) for activation and the resulting spores in each tube were inoculated into the fermentation medium (50 g). The cultures were incubated at 28 °C for 5 days until white hyphae spread over the surface of the solid medium with a small number of spores. After that, the material containing fungal cultures and extracellular enzymes was cut into 1-cm pieces, followed by air-drying (~30.0% moisture) and oven-drying (40 °C, 48 hours). The oven-dried material (~5.8% moisture) was grounded and sieved (0.10 mm).”
Comment 2)
Section 3.5.2. Author discussed results of gasifiable n-alkanes composition and propose the possible mechanisms underlying n-alkaens biodegradation by fungal enzymes in the three steps. Is the proposed mechanism based on the results of biogas composition analyses or it is just assumptions? If the results of biogas composition exist, please include results in the manuscript.
Response: The proposed mechanism about n-alkaens biodegradation by fungal enzymes in the three steps is based on GC-MS analysis of biogas composition, combined with extra-heavy oil degradation analysis (Tables 4 and 5) and organic acid analysis (Table S1). We have cited Daccò et al. (2020) and Folayan et al. (2023) to support the proposed mechanism.
Line 443, The results of biogas composition are presented in section 3.5.3. “Production of biogases and organic acids”.
Lines 423-425, “The alcohol is then oxidized to aldehyde and subsequently to fatty acid. Further, the fatty acid is broken down into CO2, H2O, non-toxic or less toxic residues via β-oxidation [14].”
Lines 425-427, “After oxygen depletion by aerobic degradation, anaerobes can transform organic compounds into useful products (e.g., CH4, acetic acid) using nitrate (NO3–), sulfate (SO42–), and carbonate (CO32–) as alternative electron acceptors [39].”
|
Figure 1. GC-MS chromatograms for gas products from extra-heavy oil within 7 days of enzymatic treatment.
Table S1
Organic acid and alcohol production from extra-heavy oil degradation by fungal enzymes of Aspergillus terreus HJ2 and A. nidulans HJ4
Retention time (S) |
E2 |
E4 |
E242 |
Derivatives |
793.9 |
212221631 |
679325920 |
138484906 |
Ethanol, TMS derivative |
812.5 |
13130808 |
16742428 |
8344425 |
Propanoic acid, 2-oxo-3-(trimethylsilyl)-, trimethylsilyl ester |
876.6 |
268595097 |
253900405 |
87143973 |
Butanoic acid, 2-amino-4,4,4-trifluoro-3-oxo-, methyl ester |
920.3 |
231958076 |
107819599 |
59091385 |
Perfluoropropionic acid, TMS derivative |
1154.5 |
14714207 |
41119832 |
6639682 |
Acetic acid, hydroxy-, ethyl ester |
1156.9 |
108259948 |
41119832 |
57699762 |
2-Keto-4-(methylthio)butyric acid, TMS derivative |
1604.4 |
54964579 |
29904022 |
17362502 |
Ethyl 2-(2,2,2-trifluoroacetamido)acetate |
1750.7 |
1997666 |
26784318 |
19913979 |
(2-Ethoxyethoxy)acetic acid, TMS derivative |
1782.3 |
59064375 |
293837096 |
319780242 |
2-Ethoxyethanol, TMS derivative |
1826.9 |
4783651 |
5912669 |
4626942 |
Malonic acid, bis (2-trimethylsilylethyl ester |
1922.1 |
29216080 |
250049 |
23952990 |
1-Hexadecanol, TMS derivative |
2005.9 |
8993571 |
1256797 |
563964 |
Methylmalonic acid, 2TMS derivative |
2124.9 |
768818 |
2450641 |
1668073 |
Malonic acid, bis (2-trimethylsilylethyl ester |
2429.5 |
52004682 |
39925994 |
40119359 |
p-Dioxane-2,3-diol |
2597.4 |
39113965 |
57511732 |
48181958 |
Erythritol, 4TMS derivative |
3364.3 |
77024914 |
52276991 |
44248133 |
Oxalic acid mono-(N-methyl-N-trifluoroacetyl)-amide, methyl ester |
3503.1 |
3704220 |
3817750 |
3595849 |
Pentanoic acid |
3626.7 |
6243780 |
340560 |
458200 |
Ethylmalonic acid, 2TMS derivative |
3875.3 |
2293204 |
20967695 |
20598585 |
Ribitol, 5TMS derivative |
E2 and E4 are fungal enzymes from A. terreus HJ2 and A. nidulans HJ4, respectively. E242 is the enzymatic consortium of E2 and E4 extracts (5:3, v/v).
REFERENCES
Daccò, C.; Girometta, C.; Asemoloye, M.D.; Carpani, G.; Picco, A.M.; Tosi, S. Key fungal degradation patterns, enzymes and their applications for the removal of aliphatic hydrocarbons in polluted soils: A review. Int. Biodeter. Biodegr. 2020, 147, 104866. https://doi.org/10.1016/j.ibiod.2019.104866.
Folayan, A.J.; Dosunmu, A.; Oriji, B. Aerobic and anaerobic biodegradation of synthetic drilling fluids in marine deep-water offshore environments: process variables and empirical investigations. Energy Rep. 2023, 9, 2153–2168. https://doi.org/10.1016/j.egyr.2023.01.034.
Comment 3)
Section 3.5.4. Please, indicate references with evidences of fungal enzyme demetallization. Mycoremediation is process used for the heavy metals depilation. How fungal enzymes depilate heavy metals in relative significant amount?
Response: We have indicated references with evidences of fungal enzyme demetallization and the mechanism by which fungal enzymes depilate heavy metals as suggested.
Lines 87-91, Added, “As for the biodegradation of petroleum hydrocarbons, heme-chloroperoxidase from Caldariomyces fumago promotes oxidation of a petroporphyrin-rich fraction in asphaltenes in the presence of hydrogen peroxide, accompanied by efficient removal of nickel (Ni) and vanadium (V) [17].”
Fungal enzymes are able to locate on the extracellular membrane launches a preliminary attack on asphaltenes at the surface of cell, destroying the petroporphyrin structure and releasing vanadium and nickel.
Lines 538-544, Added, “Nzila and Musa [49] indicated that microbes may degrade the aliphatic and aromatic moieties of asphaltene compounds, leaving the most complex parts as intermediates; thus, biomineralization of asphaltenes still remains unresolved. Nevertheless, other researchers argued that some enzymes secreted by fungi, such as the cytochrome P450 and lignin-degrading enzymes, are able to locate on the extracellular membrane and launch a preliminary attack to asphaltenes at the cell surface, destroying the porphyrin structure and releasing V and Ni [18,50].”
REFERENCES
Ayala, M.; Verdin, J.; Vazquez-Duhalt, R. The prospects for peroxidase-based biorefining of petroleum fuels. Biocatal. Biotransfor. 2007, 25, 114–129. https://doi.org/10.1080/10242420701379015.
Naranjo-Briceño, L.; Pernía, B.; Guerra, M.; Demey, J.R.; De Sisto, A.; Inojosa, Y.; González, M.; Fusella, E.; Freites, M.; Yegres, F. Potential role of oxidative exoenzymes of the extremophilic fungus Pestalotiopsis palmarum BM-04 in biotransformation of extra-heavy crude oil. Microb. Biotechnol. 2013, 6, 720–730. https://doi.org/10.1111/1751-7915.12067.
Nzila, A.; Musa, M.M. Current knowledge and future challenges on bacterial degradation of the highly complex petroleum products asphaltenes and resins. Front. Env. Sci. 2021, 9, 779644. https://doi.org/10.3389/fenvs.2021.779644.
Hernández-López, E.L.; Perezgasga, L.; Huerta-Saquero, A.; Mouriño-Pérez, R., Vazquez-Duhaltet, R. Biotransformation of petroleum asphaltenes and high molecular weight polycyclic aromatic hydrocarbons by Neosartorya fischeri. Environ. Sci. Pollut. Res. 2016, 23, 10773–10784. https://doi.org/10.1007/s11356-016-6277-1.
Comment 4)
If authors measured, composition of fungal enzymes or at least concentration of proteins in the prepared fungal enzymes mixture will be welcome to the manuscript results. Potential enzymes for the analyses (activity determination) are: lacasses, MnP...
Response: Unfortunately, we did not measure the protein content in the enzyme preparations. We agree that it is crucial to identify the enzyme members in the crude extracts for previse application in practice. We did measure dehydrogenase and catechol 2,3-dioxygenase activities in the crude extracts previously (Zhang et al. 2021). In the current study, we also preliminarily measured dehydrogenase and catechol 2,3-dioxygenase activities in the crude extracts based on color reaction, although we did not present the data.
We have mentioned the two potential enzymes in the Introduction section, Lines 98-101, “Previously, two hydrocarbon-degrading Apergillus strains, A. terreus and A. nidulans, were isolated from a bitumen sample. Their extracellular enzymes exhibit dehydrogenase and catechol 2,3-dioxygenase activities, which can reduce oil viscosity and convert oil fractions in HCO (Zhang et al., 2021).”
We also linked the enzyme activities to hydrogen production in the Discussion section. Lines 321-326, “As there were no live microbes in our reaction system, the formation of biogases, particularly H2, was most likely attributed to fungal dehydrogenase activity [19]. In ad-dition to dehydrogenases, hydrogenases have been reported as important metalloenzymes for H2 evolution and CO2 reduction in nature [31,32]. Identifying the enzymes (or proteins) present in the crude enzyme extracts can give more information on the mechanisms of biogas formation during ECHO degradation.”
Lacasses and manganese peroxidases (MnP) are two important groups of enzymes associated with petroleum hydrocarbon degradation (Rezaei and Moghimi 2024). In this study, our test results were undesirable due to multiple reasons. We will consider improving the analysis methods and assaying the activity of lacasses and MnP in subsequent studies.
REFERENCES
Zhang, J.H.; Gao, H.; Lai, H.X; Hu, S.B.; Xue, Q.H. Biodegradation of heavy oil by fungal extracellular enzymes from Aspergillus spp. shows potential to enhance oil recovery. AIChE J. 2021, 67(5), 1–11. https://doi.org/10.1002/aic.17222.
Rezaei, Z.; Moghimi, H. Fungal-bacterial consortia: A promising strategy for the removal of
petroleum hydrocarbons. Ecotox. Environ. Safe. 2024, 280, 116543. https://doi.org/10.1016/j.ecoenv.2024.116543.

Reviewer 2 Report
Comments and Suggestions for Authors
The peer-reviewed article is an original experimental study devoted to the topical and important problem of extra-heavy crude oil degradation using microbial activity. The fact that the authors determined the biodegrading activity of microscopic fungi on extra-heavy crude oil deserves special attention, which is the first such study. The title of the article clearly reflects the presented research. The aim and task are also clearly outlined. Highly appreciating the reviewed work, I would like to make some comments:
1. In the abstract and in the text, the wording regarding the production of biogas and organic acids by extracellular enzymes should be changed. The authors' formulation is nonsense, since enzymes are not capable of producing, but only contribute to the decomposition of some compounds with the formation of others.
2. "Enzymatic Consortium" is also used incorrectly in the text of the article. The test should be rewritten, not using this wording, but use, for example, "enzymatic complex". Also, the question arises, were the enzymatic preparations devoid of mycelium, fungal spores? Were the preparations sterile?
3. I also consider the definition of measurement units to be incorrect: ml×bottle-1. For comparability of results obtained by other authors earlier or in the future, for example, ml×100 ml-1 should be used.
4. Line 426: The charge of nitrate ions should be written NO3- (NO31- is not usually written this way). Also, please correct the word “sulfide” to “sulfate”, because it is this that is used as an electron acceptor. The formula of the sulfate ion is written correctly.
5. Lines 475-476: Degradation can also produce nothing. As a result of degradation, products are formed.
6. Prospects of corrosion damage to tanks during the formation of biogas and organic acids should be outlined.
7. The conclusion needs corrections according to the above-mentioned remarks (see the article file).
After the article is rewritten according to the above comments, it can be published.

Author Response
Reviewer #2
Comments and Suggestions for Authors
The peer-reviewed article is an original experimental study devoted to the topical and important problem of extra-heavy crude oil degradation using microbial activity. The fact that the authors determined the biodegrading activity of microscopic fungi on extra-heavy crude oil deserves special attention, which is the first such study. The title of the article clearly reflects the presented research. The aim and task are also clearly outlined. Highly appreciating the reviewed work, I would like to make some comments:
Comment 1)
Abstract: Extracellular enzymes from each strain separately, as well as mixtures of enzymes...
Response: We have modified the description of enzymes in the abstract and the text as suggested. For example,
Lines 28-31, Revised, “Extracellular enzymes from each strain separately, as well as mixtures of enzymes exhibited EHCO degradation activity, leading to redistribution of hydrocarbons with substantial formation of biogases and organic acids in a 7-day period.”
Original, “Extracellular enzymes from single or multiple strains exhibited EHCO degradation activity, leading to redistribution of hydrocarbons and production of biogases and organic acids within a 7-day period.”
Comment 2)
In the abstract and in the text, the wording regarding the production of biogas and organic acids by extracellular enzymes should be changed. The authors' formulation is nonsense, since enzymes are not capable of producing, but only contribute to the decomposition of some compounds with the formation of others.
Response: We have reworded the description regarding the production (changed to formation) of biogases and organic acids by extracellular enzymes as suggested.
Lines 28-31, Revised, “Extracellular enzymes from each strain separately, as well as mixtures of enzymes exhibited EHCO degradation activity, leading to redistribution of hydrocarbons with substantial formation of biogases and organic acids in a 7-day period.”
Original, “Extracellular enzymes from single or multiple strains exhibited EHCO degradation activity, leading to redistribution of hydrocarbons and production of biogases and organic acids within a 7-day period.”
Comment 3)
"Enzymatic Consortium" is also used incorrectly in the text of the article. The test should be rewritten, not using this wording, but use, for example, "enzymatic complex". Also, the question arises, were the enzymatic preparations devoid of mycelium, fungal spores? Were the preparations sterile?
Response: We have replaced “Enzymatic Consortium” with “enzymatic complex” as suggested. This has been corrected in Line 174, Revised, “Enzymatic complex optimization” and other section in the text.
The procedure used to obtain enzymatic preparations has been indicated with more detail in Lines 142-152, “Fungal enzyme preparations (mixtures of extracellular enzymes) were formulated by solid-state fermentation with the optimal carbon and nitrogen sources [19]. The fermentation medium (pH 7.0) was prepared with MSS (containing 0.4 g L–1 soluble starch and 3 g L–1 yeast extract), wheat bran, and EHCO (80:100:2, v/w/w). Two fungal strains (HJ2 and HJ4) were separately transferred to fresh PDA slants (4 mL) for activation and the resulting spores in each tube were inoculated into the fermentation medium (50 g). The cultures were incubated at 28 °C for 5 days until white hyphae spread over the surface of the solid medium with a small number of spores. After that, the material containing fungal cultures and extracellular enzymes was cut into 1-cm pieces, followed by air-drying (~30.0% moisture) and oven-drying (40 °C, 48 hours). The oven-dried material (~5.8% moisture) was grounded and sieved (0.10 mm).”
The preparations were not sterile, but the materials were oven-dried at 40 °C for 48 h, which could kill the mycelium. Additionally, the enzyme extracts were filtered through sterile fiberglass membrane filters before use, which removed most of the spores and undecomposed wheat bran. Moreover, the enzymatic degradation test was conducted in 250-mL glass bottles containing 2 g of extra-heavy oil and 40 mL of crude enzyme extract. The bottles were sealed with a rubber stopper and kept at 40 °C for 7 days. As the degradation reaction was undertaken in sealed bottles under oxygen-limited conditions, and no colonies formed on fresh PDA plates inoculated with the 7-day reaction solution, we consider that extracellular enzymes played a leading role in the degradation reaction.
Comment 4)
In my opinion, it is better to write "enzymatic complex". And in the text of the article further on as well.
Response: We have replaced “Enzymatic Consortium” with “enzymatic complex” throughout the text as suggested. The “consortium or consortia” in the text of the article is only used for bacteria and archaea, not for enzymes.
Line 174, Revised, “Enzymatic Consortium Optimization” changed to “Enzymatic Complex Optimization”.
Line 176, 178, 181, 183, 190, 331, 333, 345, 347, 369, 386, 523, 557, 566, 570, 595, Revised, “enzymatic consortia” changed to “enzymatic complexes”.
Comment 5)
I also consider the definition of measurement units to be incorrect: ml×bottle-1. For comparability of results obtained by other authors earlier or in the future, for example, ml×100 ml-1 should be used.
Response: We have corrected the measurement units as suggested.
Line 183, mL bottle-1 corrected to mL 100 mL−1 in Table 1.
Line 300, mL bottle-1 corrected to mL 100 mL−1 in Table 3.
Lines 317-318, Revised, “The total gas production with E2 and E4 was 34.8 and 27.2 mL 100 mL−1”
Original, “The total gas production with E2 and E4 was 87.0 and 68.0 mL bottle−1, respectively (Table 3).”
Lines 335-337, Revised, “the total gas yield trended upward from 48.4 to 129.0 mL 100 mL-1 (E2) and from 41.6 to 126.4 mL100 mL-1 (E4).”
Original, “the total gas production exhibited an upward trend from 121.0 to 322.5 mL 332
bottle−1 (E2) and from 104.0 to 316.0 mL bottle–1 (E4).”
Lines 449-450, Revised, “Under the optimized conditions, the total gas yield was 113.6 mL 100 mL−1 (E2), 109.2 mL 100 mL−1 (E4), and 135.6 mL 100 mL−1 (E242; Table 4).”
Original, “the total gas production was 284.0 mL bottle−1 (E2), 273.0 mL bottle−1 (E4), and 339.0 mL bottle−1 (E242; Table 4).”
Comment 6)
Line 426: The charge of nitrate ions should be written NO3- (NO31- is not usually written this way). Also, please correct the word “sulfide” to “sulfate”, because it is this that is used as an electron acceptor. The formula of the sulfate ion is written correctly.
Response: Corrected as suggested.
Line 427, Revised, “(NO3–), sulfate (SO42–), and carbonate (CO32–) as …”.
Original, “(NO31–), sulfide (SO42–), and carbonate (CO32–) as…”.
Comment 7)
Lines 475-476: Degradation can also produce nothing. As a result of degradation, products are formed.
Response: We have reworded “production of biogases and organic acids” to “formation of biogases and organic acids” throughout the manuscript. We also replaced “total gas production” and “biogas production” to “total gas yield” and “biogas yield”. For example,
Lines 475-477, Revised, “EHCO degradation by fungal enzymes was associated with substantial formation of acids, as evidenced by a notable decrease in solution pH compared to the control group.”
Original, “The EHCO degradation by fungal enzymes produced considerable acids, as demonstrated by a significant decrease in solution pH compared to the control group.”
Lines 450-452, “Compared to E2 and E4, the enzymatic complex E242 exhibited a higher ability to degrade EHCO with formation of more biogases, which could offer greater benefits to EHCO recovery.”
Original, “The enzymatic consortium E242 exhibited a greater ability to produce biogases compared to E2 and E4, thereby offering more benefits to EHCO recovery.”
Lines 461-463, Revised, “The fungal enzymes from each Aspergillus strain separately, as well as a mixture of enzymes carried out the transformation of EHCO accompanied by substantial formation of biogases (Table 4).”
Original, “The fungal enzymes from single and multiple Aspergillus strains exhibited high abilities to produce gases from EHCO (Table 4).”
Lines 465-466, Revised, “The total gas yield during HCO and EHCO degradation by E2 and E4 is higher than previously reported for bacteria [35,41].”
Original, “During HCO and EHCO degradation, both E2 and E4 demonstrated higher abilities to produce gases than previously reported bacteria [35,41].”
Lines 470-472, Revised, “A total of 335.3 mL of gases (mainly CO2 and H2) were formed by degrading 2.00 g of EHCO with E242 in a 7-day period, unlocking the potential of this enzymatic complex for application in EHCO recovery.”
Original, “The enzymatic consortium E242 produced a total of 335.3 mL of gases (mainly CO2 and H2) by degrading 2.00 g of EHCO within a 7-day period. Given its prominent gas production ability, E242 has great application potential in EHCO recovery.”
Lines 486-488, Revised, “These findings allow us to posit that organic acids were formed due to the enzymatic degradation of EHCO, which in turn affected the pH of reaction solutions.”
Original, “These findings allow us to posit that the enzymatic degradation of EHCO contributed to the decrease in solution pH through the production of organic acids.”
Comment 8)
Prospects of corrosion damage to tanks during the formation of biogas and organic acids should be outlined.
Response: We have outlined the prospects of corrosion damage to tanks during the formation of biogas and organic acids as suggested.
Lines 500-509, Added, “Trapped oil can be liberated by microbial metabolites: gases that increase pressure to force more oil out of wells and organic acids that dissolve carbonaceous deposits to increase well permeability. However, Hillman et al. [43] and Madirisha et al. [44] proposed that microbial activity in oil wells may also be disruptive to oil recovery and/or alter oil composition. For example, microbes can degrade and sour oil through specific metabolites, such as H2S, formic acid, acetic acid, and methyl alcohol. These microbial metabolites are likely to corrode the well casing, flowlines, and pipelines, despite rarely reaching particularly low pH values. The potential threat arising the formation of undesirable metabolites needs to be carefully addressed and investigated.”
Comment 9)
Enzymes for the production of EHCO?
Response: We have corrected the statement as suggested.
Lines 599-600, Revised, “Findings of this study imply that fungal enzymes from biodegradative strains can be used for exploitation and downstream processing of EHCO.”
Original, “Findings of this study imply that fungal enzymes from biodegradable strains can be used as candidates for production and downstream processing of EHCO.”
Comment 10)
The conclusion needs corrections according to the above-mentioned remarks (see the article file).
Response: We have corrected the conclusion as suggested.
Line 602, Deleted, “The addition of appropriate biosurfactants at optimal rates may enhance the biodegradation efficiency of fungal enzymes.”
Line 595, Revised, “enzymatic consortia” changed to “enzymatic complex”.
Line 594, Revised, “production of biogases and organic acids” changed to “formation of biogases and organic acids”.
Comment 11)
References, here and then highlight the year of publication in bold.
Response: We have highlighted the year of publication in bold as suggested and corrected other minor errors for consistency.
For example,
Line 626, Revised, “1. Geoenergy Sci. Eng. 2022, 222, 211409.”
Original, “Geoenergy Sci. Eng. 2022, 222, 211409.”
Line 629, Revised, “2. Biodegradation 2012, 23, 15–24.”
Original, “Biodegradation 2012, 23, 15–24.”
- Line 647, Revised, “9. biosurfactant production”
- Original, “biosurfactantproduction”
- Line 693, Revised, “26. Gulf of Mexico”.
- Original, “gulf of Mexico”.
Line 758, Revised, “51. Chem. Eng. J. 2023, 470, 144220.”
Original, “Chem. Eng. J. 2023, 470, 144220”

Round 2
Reviewer 2 Report
Comments and Suggestions for Authors
The previous comments were taken into account in the article. The article can be published